

# Biological enhancement of mineral weathering by *Pinus sylvestris* seedlings - effects of plants, ectomycorrhizal fungi, and elevated CO$_2$.

Nicholas P. Rosenstock[1,2], Patrick A.W. van Hees[3,4], Petra M.A. Fransson[1], Roger D. Finlay[1], Anna Rosling[1,5]

[1]Department of Forest Mycology and Plant Pathology, Uppsala BioCenter, Swedish University of Agricultural Sciences, Uppsala, 750 07, Sweden
[2]Present address: Center for Environmental and Climate Research, Lund University, Lund, 232 62, Sweden
[3]Man-Technology-Environment Research Centre, Örebro University, Örebro, 701 82, Sweden
[4]Present address: Eurofins Environment Sweden AB, Lidköping, 531 17, Sweden
[5]Present address: Department of Ecology and Genetics, Uppsala University, Uppsala, 752 36, Sweden

*Correspondence to*: Nicholas P. Rosenstock (rosenstockn@gmail.com)

**Abstract.** Better understanding and quantifying the relative influence of plants, associated mycorrhizal fungi, and abiotic factors such as elevated CO$_2$ on biotic weathering is essential to constraining weathering estimates. We employed a column microcosm system to examine the effects of elevated CO$_2$ and *Pinus sylvestris* seedlings, with or without the ectomycorrhizal fungi *Piloderma fallax* and *Suillus variegatus*, on rhizosphere soil solution concentrations of low molecular weight organic acids (LMWOA) and weathering of primary minerals. Seedlings significantly
increased mineral weathering, as estimated from elemental budgets of Ca, K, Mg, and Si. Elevated CO$_2$ increased plant growth and LMWOA concentrations, but had no effect on weathering. Colonization by ectomycorrhizal fungi, particularly *P. fallax,* showed some tendency to increase weathering. LMWOA concentrations correlated with seedling biomass across both CO$_2$ and mycorrhizal treatments, but not with total weathering. We conclude that nutrient uptake, which reduces transport limitation to weathering, is the primary mechanism by which plants enhanced weathering in
this system. While the experimental system used departs from conditions in forest soils in a number of ways, these results are in line with weathering studies performed at the ecosystem, macrocosm, and microcosm scale, indicating that nutrient uptake by plants and microbes is an important biological mechanism by which mineral weathering is enhanced.




## 1 Introduction

Accurate estimates of net primary productivity of forests are critically important to global carbon models, and forest productivity is predicted to increase due to elevated $CO_2$ (Lindner *et al.* 2010; Ainsworth and Long 2005). The extent of this response to elevated $CO_2$ levels is largely dependent on the ability of forest trees to meet their increased carbon

availability and water use efficiency with increased nutrient uptake (Pinkard *et al.* 2010; Norby *et al.* 1999). As the effects of anthropogenic nitrogen deposition continue to accumulate, large areas of forest are limited by mineral-derived nutrients (Naples and Fisk 2010; Baribault *et al.* 2010; Jonard *et al* 2015). The ability of forest ecosystems of replenishing pools of base cations and phosphorus through dissolution of minerals may thus become increasingly important. In coniferous trees, elevated $CO_2$ has been shown to increase the ratio of root to shoot biomass (Alberton *et*

*al.* 2007; Janssens *et al.* 2005) and allocation to mycorrhizal symbionts (Fransson *et al.* 2010; Compant *et al.* 2010) indicating an increased exploration of the soil environment to obtain nutrients. Studies in Swedish (Almeida *et al.* 2018) and Czech (Rosenstock *et al.*, 2016) Norway spruce forests have observed increased belowground allocation to fine roots and ectomycorrhizal hyphal growth and greater preference for growth around apatite (a mineral source of Ca and P) mineral grains under phosphorus-limiting conditions.

Most forest trees of the temperate and boreal biomes are dependent on ectomycorrhizal fungi for their survival (Smith and Read, 2008). Ectomycorrhizal fungi (EMF) are mutualistic symbionts that form intimate associations with the fine roots of trees and some woody shrubs, and colonise mineral and organic substrates in the soil with their extraradical mycelia. Increased nutrient uptake is generally considered to be the most beneficial effect of EMF on forest trees (Smith and Read, 2008), and the host plant transfers significant amounts of photosynthetically fixed carbon

to their associated EMF. A review by Hobbie (2006) suggests that an average of approximately 15% of total fixed carbon is allocated to EMF symbionts, but some studies show that more than 60% of recent carbon assimilation (Rosling *et al.*, 2004) and net primary production (Godbold *et al.*, 2006) can be allocated to EMF. As a result of their central role in tree nutrition and their ability to stimulate mineral weathering (Finlay *et al.*, 2009; 2019), EMF are acknowledged to be instrumental in forest biogeochemistry. Many studies have found that EMF influence weathering,

and the proposed mechanisms include acidification (Balogh-Brunstad *et al.*, 2008; Rosling *et al.*, 2004), nutrient uptake (Wallander, 2000; van Hees *et al.*, 2004), production of siderophores (Ochs *et al.*, 1993; Watteau and Berthelin, 1994), production of low molecular weight organic acids (Paris *et al.*, 1996, van Schöll *et al.*, 2006; Schmalenberger *et al.*, 2015 ) and biomechanical forcing (Bonneville *et al.*, 2009; 2016). Despite numerous potential mechanisms by which EMF may directly influence weathering rates, little is known about their quantitative importance for the process of

mineral weathering in soil.

In this experiment we use a column microcosm system to quantify nutrient uptake by Scots pine seedlings and determine the effect of their associated EMF, elevated $CO_2$ and soil solution LMWOA concentrations on system nutrient budgets. The experiment was designed to test the hypothesis that elevated $CO_2$ would drive plants to allocate more carbon below ground, to roots, EMF, and exudation, thereby increasing mineral weathering through quantitative

and qualitative effects on nutrient release and acquisition from the mineral substrate. We specifically asked the questions; 1. What is the contribution of plants compared to EMF to mineral weathering?, 2. Will elevated $CO_2$ increase mineral weathering?, 3. How are LMWOA concentrations affected by mycorrhizal colonization and elevated $CO_2$?, and 4. If observed, what are the potential mechanisms of biological enhancement of weathering?



## 2 Methods

### 2.1 Experimental overview

We used column microcosm systems filled with a mineral mix mimicking a podzol E-horizon. Treatments were factorial with +/− $CO_2$, +/− seedling, and +/− EMF (two species). Non-mycorrhizal and non-planted treatments were

performed in four replicates and EMF treatments in five; in total there were 36 columns, 28 of which were planted with six-month-old seedlings. A bacterial suspension was added to all the columns to enable bacterial-mycorrhizal interactions. Following planting, the experiment was run for nine months and watered with nutrient solution. Organic acid concentrations were measured in rhizosphere soil solution, and elemental concentrations and pH were measured in column leachate collected during the experiment. Upon harvest, the ion-exchangeable element pool (EC) of the mineral

mix was quantified pre- and post- nine months of incubation in columns, in addition to the dry weight and elemental contents of seedlings, and the chitin contents of the roots and mineral mix were. Amounts of elements added with nutrient solution, taken up by plants and leached though the columns were used to construct whole column elemental budgets for silica, calcium, potassium, and magnesium.

### 2.2 Plant and mycorrhizal pre-culture and bacterial inoculation

Ectomycorrhizal and non-mycorrhizal *Pinus sylvestris* L. Karst. seedlings were prepared in Petri dishes containing peat:vermiculite:modified Melin-Norkrans media (1:4:2 V:V:V) as detailed by Fransson and Johansson (2010). The ectomycorrhizal fungal species *Suillus variegatus* (Sw.:Fr.) O. Kuntze (isolate code UP597, GenBank accession no. EF493256) and *Piloderma fallax* (Liberta) Stalpers (UP113, DQ179125), growing on half-strength, modified Melin-Norkrans medium (Marx, 1969) were used as EMF inoculum. After twelve weeks growth (300 µmol m$^{-2}$ s$^{-1}$

photosynthetic photon flux density (PPFD); 16 h light at 18°C, 8 h dark at 15°C) seedlings were removed from the Petri dishes and planted in 10 cm x 10 cm x 10 cm pots filled with a 1:10 v/v sterilized, autoclaved peat:quartz sand mixture. These pots were maintained under the conditions detailed above for 6 months and watered three times weekly with ~ 20 mL of nutrient solution. The composition of this nutrient solution was 600 µM $NH_4NO_3$, 140 µM $K_2HP0_4$, 150 µM $Ca(NO_3)_2$, 80 µM $K_2SO_4$, 15 µM $H_3BO_3$, 0.3 µM $Na_2MoO_4$, 0.3 µM $ZnSO_4$, 0.3 µM $CuSO_4$, 50 µM

$Mg(NO_3)_2$.

      In order to incorporate bacterial–ectomycorrhizal interactions, which may be important to mineral weathering (Uroz *et al*., 2007; 2009), experimental columns were inoculated with a fungus-free bacterial suspension extracted in April 2008, from E horizon soil and humus collected from a local *P. sylvestris* boreal forest (59.785N, 17.683E, Lunsen forest, Uppsala, Sweden). The collected soil was subjected to sequential centrifugation in Winogradsky salt solution

(Faegri *et al*., 1977), followed by Nycodenz (Medinor AB, Stockholm, Sweden) extraction at ultra-high speed (Courtois *et al*., 2001). The resulting bacterial suspension was tested for the presence of culturable fungi by plating on potato dextrose agar, which yielded no observable fungal colonies. The bacterial suspension was used for inoculation within two days of extraction and was stored at 4-8°C until applied.

### 2.3 Soil column system and growth conditions

The pre-grown seedlings were transplanted from pots into a sand culture system with opaque plexiglas tubes (⌀ 4 cm, height 30 cm) serving as vertical growth columns (van Hees *et al* 2006). Each column was filled with 405 grams (dry weight) of a mineral mix (which mimicked the e-horizon of a local boreal forest soil) comprised of 50% quartz sand, 28% oligoclase, 18% microcline, 1.8% hornblende, 0.9% vermiculite, and 0.9% biotite. The quartz sand was acid washed (10% HCl w/v) overnight and washed with deionized water until the solution pH was >6 before being mixed



with the other minerals. The complete mix was approximately 40% silt size class (70 μm - 100μm) and 60% sand size class (100μm - 500μm). The columns drained into opaque 250-mL glass bottles via a ceramic lysimeter cup (655X01, Soil Moisture Corp., Santa Barbara, CA) at the base of the columns. Suction was applied to the bottles to induce drainage. After packing the mineral mixture into the columns and before seedlings were planted into the columns, the

watering regime and drainage efficiency were established over three weeks. Rhizon SMS-MOM suction lysimeter samplers (0.3 cm diameter, 3 cm length; Rhizosphere Research Products, Wageningen) were inserted horizontally ten cm below the soil surface to extract rhizosphere soil solution and measure LMWOA concentrations within. One seedling was planted in each column (except for the 8 non-planted controls). At the time of planting three seedlings from each treatment (non-mycorrhizal, *P. fallax*, *S. variegatus*) were dried for future analysis. One week after planting,

each column was inoculated with 5 mL of the fungus-free bacterial inoculum described above.

The columns were incubated in two adjacent, climate-controlled chambers (16 h light at 20 °C, 8 h dark at 15 °C); in one chamber $CO_2$ levels were maintained at 330-380 ppm (ambient) and in the other 700-750 ppm (elevated). Light was supplied by a high-pressure sodium lamp with an intensity of 300 PPFD at the seedling tops. The columns were watered three times a week with nutrient solution (72-108 mL column$^{-1}$ week$^{-1}$). The watering solution was 33

μM $(NH_4)_2HPO_4$, 407 μM $NH_4NO_3$, 27.5 μM $K_2HP0_4$, 55 μM $Ca(NO_3)_2$, 27.5 μM $K_2SO_4$, 5.5 μM $H_3BO_3$, 1 μM $FeCl_3$, 0.1 μM $Na_2MoO_4$, 0.1 μM $ZnSO_4$, 0.1 μM $CuSO_4$, 55 μM $Mg(NO_3)_2$, and the pH was adjusted to 5.0. Four hours after watering, suction was applied for one hour to the bottom of the columns via the ceramic lysimeters and the column leachate was collected in the glass bottles beneath each column.

**2.4 Sampling and chemical analysis**

Column leachate was sampled from drainage bottles beneath each column every 3-4 weeks throughout the experiment (in total 11 times). At each sampling, solution volume was measured and duplicate 15 ml aliquots from each column were collected and frozen for future elemental analysis. pH was measured for the leachate of each column at seven of the sampling dates. Rhizosphere soil solution was extracted for low molecular weight organic acid (LMWOA) analysis by applying suction to lysimeter samplers 24–36 h after watering. Rhizosphere soil solution samples were collected

five times from each column at 5, 6, 7, 8, and 9 months post planting, and samples were immediately frozen at -20 °C for subsequent analysis.

At harvest, seedlings were removed from the mineral mix, adhering mineral particles were classified as the rhizosphere soil fraction and collected by dry-shaking the roots. Rhizosphere and bulk mineral mix fractions were collected, weighed, dried, and stored for future analysis. Seedling roots were washed with deionized water and

classified as abundantly colonized (>50% of root tips colonized), moderately colonized (50% > colonization >5%), or sparsely colonized (<5% colonized) by ectomycorrhizal fungi. Each seedling was separated into roots, stem, and needles and dried at 60 °C for 72 hours, after which dry weight (DW) was measured.

Concentrations of LMWOAs in rhizosphere soil solution samples were determined by capillary electrophoresis using the method of Dahlén *et al.* (2000). Briefly, LMWOA's were analyzed on an Agilent $^{3D}$CE

capillary electrophoresis system (Agilent Technologies, Santa Clara). The concentrations of 12 different LMWOAs were analyzed: acetate, butyrate, citrate, formate, fumarate, lactate, malate, malonate, oxalate, proprionate, succinate, and shikimate. LMWOA data is presented in μmol/L solution collected from rhizosphere lysimeters and as μmol/L/gram plant DW.

To measure elemental contents, acid digestion of plant material was undertaken following the procedure of

Zarcinas *et al*. (1987) as follows: 0.1 gram of each seedling component (needles, stems, and roots), all pre-ground on a Wiley® mill (Thomas Scientific; Swedesboro, USA) was separately digested at room temperature overnight in 2 mL



concentrated $HNO_3$ (10N), heated up to and refluxed at 130 $^\circ$C with a funnel lid for five to seven hours and subsequently diluted with 12-15 mL deionized water.

Ammonium-acetate extractable elements (EC) were measured in the post-harvest mineral mix for each column as well as for nine replicates of the pre-experimental mineral mix. Extractions were performed in a 1:10 (m:V) mineral

mix:1M $NH_4Ac$ suspension by shaking for 5 hours at 100 rpm at room temperature. The supernatant was separated by centrifugation (3000g) and filtered through a pre-washed 0.45 μm Na-acetate filter syringe. The pre-experimental mineral mix was equilibrated with the nutrient solution 3 x 12 h to mimic the pre-planting treatment of minerals in columns.

Plant digests, EC extracts, and column leachate were all analyzed for elemental contents of Al, Ca, Fe, K, Mg,

Mn, Na, P, S, and Si on a Perkin-Elmer (Waltham, USA) atomic optical emission, inductively coupled, plasma emission spectrometer (AOE-ICP). A set of four standards was established based on preliminary analysis for each sample type. In addition to hourly re-running standards, duplicates and an internal scandium standard were run to ensure an accuracy of elemental contents to +/- 1%. Total elemental loss (μmol) through column leachate was calculated from the leachate concentration and the total volume of leachate at each sampling time.

Plant roots and mineral mix were assayed for chitin content post-harvest to assess fungal biomass. Chitin was extracted and analyzed by HPLC at the Department of Forest Ecology & Management, SLU (Sweden), according to the method of Ekblad and Näsholm (1996). The chitin concentration of roots, rhizosphere mineral mix, and bulk mineral mix were multiplied by the mass of that fraction and these sums were added to obtain total chitin content per column. To relate fungal biomass to plant biomass the total chitin content (root + rhizosphere + mineral mix) was

divided by the total plant biomass in each column.

**2.5 Construction of elemental budgets to estimate weathering**

Elemental budgets of Ca, K, Mg, and Si were constructed to estimate weathering, using the four pools: 1. nutrients added ($N_a$); 2. seedling uptake ($S_u$: final elemental content in plants – pre-experimental plant contents); 3. elemental leaching ($L_t$: total leachate); 4. ammonium-acetate extractable elements ($E_C$). All values were calculated in micromoles

and minimum weathering ($W_m$) was represented by the following equation.

$$W_m = \Delta E_c + S_u + L_t - N_a$$

eq. 1

We use the term minimum weathering to account for possible secondary minerals which may have formed during the experimental period and which were not extracted with the ammonium acetate extraction. $\Delta E_c$ is the change in

ammonium acetate extractable elements over the course of the experimental period and it is calculated as:

$$\Delta E_c = E_{c-final} - E_{c-initial}$$

eq. 2

Transpiration by the seedlings was calculated as the % water lost of the total solution added as:

$$\frac{(leachate_{np} + MC_{np}) - (leachate_p + MC_p)}{3.4l} * 100\%$$

eq. 3

where $leachate_{np}$ is the total solution volume (mean mL/column) collected from column flowthrough in nonplanted treatments, $leachate_p$ is the total column flowthrough in planted treatments, $MC_{np}$ is the total solution volume remaining in nonplanted columns (dry mineral mass X % moisture determined by drying; mean mL/column) upon harvest, and



$MC_p$ is the total solution volume remaining in planted columns upon harvest. In total, 3.4 liters of nutrient solution were added to the columns over the course of the experiment.

To evaluate the potential for nutrient uptake to stimulate weathering via reducing solution concentrations of weathering

products, the effect of plants on column solution concentrations of weathering products was estimated as the portion of weathered and added elements (in nutrient solution) which plants removed from solution as:

$$\%uptake = \frac{S_u}{W_m + N_a} \times 100\%$$

eq. 4

which can also be expressed as:

$$\%uptake = \frac{S_u}{\Delta E_c + S_u + L_t} \times 100\%$$

eq. 5

## 2.6 Statistical analysis

Except where explicitly stated otherwise, all data are presented as the mean per column for each treatment +/− standard

error of the mean (SE). LMWOA and chitin are also presented as mean per column per unit seedling mass. Two different independent variables were investigated: $CO_2$ (ambient and elevated) and biological treatment (non-planted, non-mycorrhizal, *P. fallax*, *S. variegatus*). For some comparisons only the planted treatments (non-mycorrhizal, *S. bovinus* and *P. fallax*) were examined, henceforth referred to as the "planted" treatments. Two-way ANOVA was used first to determine treatment and interaction effects. When looking at the effects of the planted treatments the two $CO_2$

treatments (ambient and elevated) were combined, and when examining the effects of $CO_2$ the planted treatments (non-mycorrhizal, *P. fallax*, *S. variegatus*) were combined. Statistical analysis was performed using JMP software version 5.01a (SAS Institute, Inc., Cary, NC, USA) and R v. 3.1.2 (R Core Team, 2014). Significant differences between $CO_2$ treatments were assessed with Student's T test, and significant differences between biological treatments with Tukey's HSD test, both using a one-way ANOVA.

The chemical speciation and equilibria model Visual MINTEQ (Gustafsson, 2007) was used to examine the possibility of formation of secondary minerals within a range of chemical conditions based on data collected in our experiment. The model was run across a range of LMWOA concentrations (0 – 10 X observed), pH (pH 5 – observed) and leachate elemental concentrations (0 – 2 X observed) as model inputs.

## 3 Results

### 3.1 Seedling growth and mycorrhizal colonization

Elevated $CO_2$ significantly increased the biomass of seedlings ($P < 0.04$), but had no significant effect on root:shoot ratio.  Mycorrhizal treatment had no significant effect on growth or root:shoot ratio (Table 1). There was no interactive effect of mycorrhizal treatment and $CO_2$.

The root systems extended 12-21 cm down the 30 cm columns, but the majority of roots were densely clustered in the

top 10 cm. Mycorrhizal colonization was highly variable.  Of the 20 seedlings in the mycorrhizal treatments 6, 4, and 10 seedlings were abundantly, moderately, and sparsely colonized, respectively, at the time of harvest.   No



mycorrhizae were observed deeper than 6 cm. In the non-mycorrhizal treatment, turgid, smooth, black root tips were observed which may have been thelephoroid mycorrhiza. Chitin content in non-planted controls was negligible, and significantly higher in planted treatments (Figure 1) ($P < 0.002$). Non-mycorrhizal treatments had, on average, half as much chitin per column as the mycorrhizal treatments. Mycorrhizal treatments had significantly more chitin per gram

seedling biomass (Table 1, Figure 2), compared to the non-mycorrhizal treatment. In the *P. fallax* treatments elevated $CO_2$ was associated with significantly higher chitin per unit seedling biomass (Figure 2).

### 3.2 Low molecular weight organic acids

Formic, lactic, and acetic acids made up the majority of measured LMWOAs, comprising 82%, 12%, and 4% of total measured LMWOAs, respectively.   Much smaller amounts of malonic, oxalic, fumaric, and succinic acids were

occasionally detected, but their occurrence in measurable quantities was not associated with any specific treatment. Planted columns had significantly higher LMWOA concentrations compared to non-planted columns ($P < 0.001$), while *P. fallax* columns had significantly lower LMWOA concentrations ($P < 0.05$) than either the non-mycorrhizal or *S. variegatus* columns (Figure 3a). Formic acid concentrations and total LMWOAs were significantly higher ($P < 0.01$) in the elevated $CO_2$ treatments compared to ambient conditions (Figure 3b).

### 3.3 Seedling transpiration, leachate pH and elemental losses

Seedlings transpired on average 23% (+/− 2% SEM) of the water added to each column. Leachate pH was consistently alkaline (pH 7.0-9.4) despite the added nutrient solution being acid at pH 5. There was no significant difference in leachate pH between elevated and ambient $CO_2$ treatments or among planted treatments. On five of the seven sampling dates leachate of the non-planted controls had significantly lower pH (P < 0.001) than that of planted treatments

(Figure 4). At the first time point (43 days after planting) both planted and non-planted columns had an average pH of 8.8 (+/− 0.04 SEM). Over the course of the experiment the pH of the leachate was stable in planted columns (8.6 +/− 0.2 SEM), while the leachate from the non-planted columns decreased (~0.8 pH units), suggesting that seedlings stabilized the columns at the initial alkaline conditions. Under elevated $CO_2$, less K, Ca, Mg, and Si was lost from the columns in leachate compared to the ambient $CO_2$ treatment. More K, Ca and Mg leached from non-planted treatment

than for the planted treatments while the opposite was true for Si (Table 2). Concentrations of K, Ca, Mg, and Si in leachate were relatively steady over time (data not shown).

### 3.4 Extractable elements

Overall, $CO_2$ had no effect on EC elements . For Ca and Si, planted columns had significantly higher EC than non-planted columns, while mycorrhizal treatment, but not seedlings alone, increased the EC for Mg and K (Table 3). For

some elements and treatments the EC was considerably higher before the experiment than after (Table 3). In particular, a large decrease in EC for Ca was observed. Decreased EC was also observed for Mg and K in non-mycorhizal treatments as well as for *P. fallax* in the case of Mg.

### 3.5 Seedling elemental contents

The needle concentrations of Ca, K, Mg, Fe, and P were all at or above sufficiency thresholds for *P. sylvestris*

(Breakke, 1994; Bargagli, 1998) (data not shown). Total seedling contents of Ca, K, or Mg did not vary significantly between the $CO_2$ treatments (Table 4), despite significant differences in seedling biomass between $CO_2$ treatments (Table 1). Mycorrhizal treatment was not associated with a significant difference in total seedling contents of Ca, K, or Mg, but EMF colonization, particularly with *P. fallax*, increased seedling Mg concentration (Table 4).



### 3.6 Estimating weathering using an elemental budget

Whole column elemental budgets for the elements Si, Ca, K, and Mg show no effect of $CO_2$ on element pools (Table 5). The presence of seedlings, mycorrhizal or not, significantly enhanced weathering compared to non-planted controls (Table 5). Mycorrhizal inoculation had no significant effect on weathering for any of the examined elements.

Compared to the non-mycorrhizal and *S. variegatus* treatments more Si (23%) and Mg (27%) were weathered in columns planted with *P. fallax*-inoculated seedlings, but these differences were not statistically significant (Table 5). The major sink for weathered products Mg, K, and Ca in the seedling treatments was seedling uptake; particularly for Mg (Figure 5b-d). For Si, seedling uptake was negligible (Figure 5a). Seedlings took up ~5% of the total dissolved Si, 68% of the total dissolved K (+/-3% S.E.), 54% of the total dissolved Ca (+/-3% S.E.), and 91% of the total dissolved

Mg (+/-3 % S.E.). After subtracting the nutrients added from the elemental budgets the net weathering of Ca, K, and Mg in the non-planted treatment were negative or only slightly positive (Table 5). No formation of precipitates was predicted using the chemical speciation and equilibria model Visual MINTEQ (Gustafsson, 2007)

## 4 Discussion

### 4.1 Growth and organic acid concentrations

Elevated $CO_2$ increased the biomass of the *P. sylvestris* seedlings. Other studies of coniferous seedlings have generally found a growth stimulation with elevated $CO_2$ (reviews by Ceulemans and Mousseau, 1994; Norby *et al*., 2005), including several studies of mycorrhizal *P. sylvestris* seedlings (Gorissen and Kuyper, 2000; Alberton *et al*., 2007), though other studies have also found no plant growth effect of elevated $CO_2$ (Gorissen and Kuyper, 2000; Fransson and Johansson, 2010; Fransson *et al*., 2007). Plant growth stimulation from elevated $CO_2$ is often accompanied by an

increase in root:shoot ratio (Gorissen and Kuyper, 2000; Janssens *et al*., 2005; Alberton *et al*., 2007). While we did not detect any change in root:shoot ratio, elevated $CO_2$ was associated with significantly more chitin, in both roots and bulk mineral mix and especially for *P. fallax*, possibly indicting higher belowground allocation to fungi under elevated $CO_2$. Fransson and Johansson (2010) found *P. fallax* to be the most responsive to elevated $CO_2$ among five fungal species examined. Increased growth of mycorrhizal fungi in response to elevated $CO_2$ treatments (see reviews by Alberton *et*

*al*. 2005; Compant *et al*., 2010) is highly species-specific (Fransson *et al*. 2007; Gorissen and Kuyper, 2000; Parrent and Vilgalys, 2007). Elevated $CO_2$ was associated with significantly higher total LMWOA concentrations, potentially indicating greater carbon allocation to root exudation by seedlings under elevated $CO_2$, in agreement with previous studies (Fransson and Johansson, 2010).

### 4.2 Column nutrient budgets

After subtracting the nutrients added from the elemental budgets the net weathering of Ca, K, and Mg in the non-planted treatment are negative or only slightly positive (Table 5), suggesting a "missing sink" for weathering products in these treatments, or possibly calling into question the negative $\Delta E_C$ values. As stated previously, $\Delta E_C$ was negative for Ca and slightly negative for K and Mg in some treatments. Negative $\Delta E_C$ arises from the mineral mix having higher ammonium-acetate extractable elements at the beginning of the growth period than at the end. Alternatively, it is

possible that the equilibration with nutrient solution that occcured over 3 weeks in the columns prior to planting was not accurately replicated by the flushing steps we subjected the pre-mineral mix in preperation for measureing the EC on the "before-planting" mineral mix. We think it is likley that significant elemental losses may have occurred during this 3 week flushing phase before planting, but this would have been the same for all treatments and should thus not





affect our interpretation of biological enhancement of weathering. Alternatively, it is possible that secondary precipitates formed during the experimental period that we were not able to completely extract with the ammonium acetate extraction. Using a broad range of solution chemistry we were not able to predict the formation of precipitates using the chemical speciation and equilibria model Visual MINTEQ (Gustafsson, 2007). In addition, at the pH's we

observed, we would expect higher pH to be associated with greater secondary mineral formation, and we observed that the planted treatments had both higher pH and higher estimated total weathering.

**4.3 Biotic enhancement of weathering**

Biotic enhancement of weathering was observed in the present study as the presence of seedlings had a significant effect on every individual elemental flux. For K, Mg and Ca increased weathering products were primarily taken up by

the seedlings, while for Si increased weathering products were recovered from the leachate and mineral matrix. Elevated $CO_2$ had no significant effect on the weathering of Ca, K, Mg, or Si, despite increased plant growth in this treatment.

Soil biota are capable of directly stimulating weathering of alumina silicate minerals by four distinct mechanisms: *proton-promotion*, via biological proton exudation; *ligand-promotion*: via organic ligand exudation;

*reduction of transport limitation* via nutrient uptake; and *physical disruption* via physical forcing of minerals by hyphae or root hairs. There is considerable debate as to which of these four mechanisms dominates the biotic influence on weathering (see reviews by Drever, 1994; Hinsinger *et al*., 2006; Lucas, 2001; Harley and Gilkes, 2000 and articles by Hinsinger *et al*., 2001; Bonneville *et al*., 2009). In the present study, seedling uptake of nutrient cations, and the resulting reduction in transport-limitation to weathering, stands out as the most likely mechanism of biological

enhancement of weathering through which the presence of *P. sylvestris* caused the observed elevated weathering. Removal of transport limitation may occur if seedlings significantly reduce the concentrations of weathering products at or near mineral surfaces and/or significantly alter solution flow rates. By transpiring 23% of column moisture and taking up 54-91% of the major nutrient cations from the solution, seedlings exerted considerable influence on both column hydrology and elemental concentration gradients in this system.

In light of the alkaline pH of the leachate, proton-promotion was probably not a likely mechanism by which seedlings enhanced weathering. The high pH of the column leachate (7.0-9.4) may not reflect the pH of the rooting zone since leachate pH was collected after 30 cm of vertical percolation through ground primary minerals, while ectomycorrhize and most roots were restricted to the uppermost portions of the columns. However, the pH from planted treatments was significantly higher than for unplanted columns and remained high while unplanted columns

became increasingly neutral over the course of the experiment. This trend is the opposite of what would be expected were biotic acidification (proton-promoted dissolution) to be an important mechanism by which seedlings enhanced weathering in this system.

Ligand-promoted dissolution appears unlikley to have contributed significantly to biological enhancement of weathering, since concentrations of the most weathering-enhancing organic acids were in the order of micromolar

concentrations in the present study, not in the required millimolar concentrations to significantly increase weathering (Drever and Stillings, 1997; Drever, 1994; Pokrovsky *et al*., 2009). The stimulatory effect of LMWOA's on mineral dissolution increases significantly at near-neutral and slightly alkaline pH's, as the relative rate of proton-promoted dissolution drops sharply (Welch and Ullman, 1993; van Hees *et al*., 2002; Stillings *et al*., 1996). However, the majority of LMWOA's detected in this experiment were the mono-carboxylic acids formic, lactic, and acetic acids, not

the di-and tri-carboxylic acids such as citric and oxalic acids that have been shown to strongly increase weathering rates (Neaman *et al*., 2006; Drever and Stillings, 1997).



In soil systems LMWOA concentrations are a result of production and microbial uptake rates as well as adsorption (van Hees *et al*, 2005). LMWOA concentrations may have been considerably higher near mineral surfaces and under biofilms than those we measured, however the rhisosphere samplers were placed in the zone of highest root activity. LMWOA's represent one of a few possible weathering-promoting ligands of biotic origin, and commonly
comprise less than 10 % of dissolved organic carbon (Strobel, 2001). Organic compounds that are generally found in much lower concentrations than LMWOA's may be key ligand-promoted weathering agents in soil, such as siderophores (Lierman *et al*., 2000; Reichard *et al*., 2007; Watteau and Berthelin, 1994) which are exuded by EMF (van Hees *et al*., 2006) and bacteria (Liermann *et al*., 2000). We did not find detectable levels of iron or aluminum in column leachate (< 0.1μM [Al] and [Fe] vs. > 50μM [Si], data not shown), and thus, siderophores were likely not a
significant contributor to biotic weathering enhancement in our system.

The findings presented here are in line with a growing acceptance of nutrient uptake as a biological driver of mineral weathering. Experiments using column reactors that, unlike batch-reactors, are not saturated and well-mixed, have shown that element transport away from mineral surfaces is the rate limiting step for weathering (Evans and Banwart, 2006; van Grinsven and van Riemsdijk, 1991). Our results are thus in line with a growing body of evidence
that identify transport limitation as the predominant process governing weathering rates of minerals in soil (Harley and Gilkes, 2000; Maher, 2010). In a review of ectomycorrhizal weathering studies, Rosenstock (2010) also identified increased nutrient uptake as the main mechanism by which ectomycorrhizal fungi stimulate weathering. By taking up nutrients and stimulating solution flow, plant growth exerts a major influence on solution composition around mineral surfaces. While the actual weathering mechanism is likely ligand- or proton-promotion, the weathering rate is
controlled by nutrient uptake by plants and soil microorganisms.

## 5 Conclusions

In this experiment seedlings significantly increased weathering rates, while ectomycorrhizal colonization had limited effects; *P. fallax* showed some potential to enhance weathering rates, particularly with respect to Mg. The lack of a mycorrhizal effect may have been due to low mycorrhizal colonization and a lack of mycorrhizal plant growth
promotion. Enhancement of weathering by seedling growth was most likely explained by nutrient uptake through its influence on transport-limitation. Elevated $CO_2$ significantly increased plant growth and soil solution LMWOAs concentrations, but did not enhance total weathering. Caution is advised when relating results from ectomycorrhizal weathering experiments in containers with primary minerals and low organic content to phenomenon occurring in natural forests. Studies on biotic weathering should separate biotic influences on weathering mechanisms (proton
promotion, ligand-promotion) from biotic influences on weathering rate-limiting factors (transport limitation).

## Author contribution

PvH, AR, PF, RF, and NR designed the experiment. AR, PF, and NR conducted the experiment. NR analyzed the data. NR prepared the manuscript with assistance and contributions from AR. PF. PvH, and RF.


## Competing interests

The authors declare that they have no conflict of interest.



**Acknowledgements**

This work was made possible thought funding from the Swedish Research Council. We thank Torgny Näsholm for performing the chitin analysis, and Paul Brooks for assistance with ICP analysis.

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




|  | Ambient CO$_2$ | Elevated CO$_2$ | Non-mycorrhizal | *Suillus variegatus* | *Piloderma fallax* |
|---|---|---|---|---|---|
| Total biomass (g) | 3.84 (0.21)a | 4.49 (0.21)b | 4.67 (0.24)a | 4.05 (0.24)a | 3.88 (0.29)a |
| Root:shoot | 1.98 (0.17)a | 1.81 (0.12)a | 2.16 (0.18)a | 1.86 (0.20)a | 1.71 (0.14)a |
| Seedling chitin (mg/g) | 12.3 (2.19)a | 15.9 (2.90)a | 7.1 (0.99)a | 17.7 (3.11)b | 16.2 (3.36)b |

**Table 1: Seedling biomass, root:shoot mass ratio, and seedling chitin contents (mg chitin/ g seedling dry weight) by treatment (mean per column). Values in parentheses are standard errors of the mean; values paired with different letters are significantly different ($p < 0.05$) either between ambient and elevated CO$_2$ (left half of table) or among planted treatments (right half of table).**

| Element | Ambient CO$_2$ | Elevated CO$_2$ | Non-planted | Non-mycorrhizal | *Suillus variegatus* | *Piloderma fallax* |
|---|---|---|---|---|---|---|
| Si | 166 (7.5) b | 134 (4.7) a | 103 (9.5) a | 154 (8.6) b | 151(10.3)b | 145 (9.1) b |
| Ca | 467 (15.5)b | 370 (14.6)a | 588 (17.0)b | 431 (35.4)a | 417 (16.5)b | 410 (24.3)a |
| K | 220 (9.7)b | 163 (10.7)a | 416 (14.8)b | 183 (16.2)a | 181 (11.9)a | 209 (17.8)a |
| Mg | 43.0 (1.9) b | 31.7 (1.6) a | 61.8 (2.4)b | 36.5 (2.8)a | 36.1 (2.3)a | 39.4 (3.5)a |

**Table 2: Mean total elemental losses in column leachate (total µmol/ column). Values in parentheses are standard errors of the mean; values paired with different letters are significantly different ($p < 0.05$) either between planted columns with ambient and elevated CO$_2$ (left half of table) or between biological treatments (right half of table).**

25

| Element | Ambient CO$_2$ | Elevated CO$_2$ | Non-planted | Non-mycorrhizal | *Suillus variegatus* | *Piloderma fallax* |
|---|---|---|---|---|---|---|
| Si | 146 (46.3)a | 179 (42.3)a | 38 (13.0)a | 141 (52.5)b | 120. (49.2)b | 222. (57.1)b |
| Ca | -515 (56.9)a | -500 (10.8)a | -646 (8.2)a | -514 (19.4)b | -504 (17.8)b | -507 (12.7)b |
| K | 26.3 (11.7)a | 34.3 (10.4)a | -44.1 (7.7)a | -4 (12.7)a | 50.1 (13.1)b | 38 (8.6)b |
| Mg | -28.0 (7.5) a | -24.1 (6.4) a | -59 (5.4)a | -36 (7.7) a | 1.8 (7.5) b | -10.1 (5.5) b |

**Table 3: Change in surface ammonium-acetate extractable elements ΔEc (Ec$_{final}$ − Ec$_{initial}$: mean µmol/ column). Values paired with different letters are significantly different ($p < 0.05$) either between planted columns with ambient and elevated CO$_2$ (left half of table) or biological treatments (right half of table).**

30





| Element | Ambient $CO_2$ | Elevated $CO_2$ | Non-mycorrhizal | *Suillus variegatus* | *Piloderma fallax* |
|---|---|---|---|---|---|
| Si (μmol/seedling) | 13.7 (0.93)a | 17.5 (1.49)b | 16.0 (1.61)a | 15.0 (1.15)a | 15.9 (2.08)a |
| Ca (μmol/seedling) | 651 (49.8)a | 757 (45.5)a | 776 (60.7)a | 740. (54.3)a | 611 (57.2)a |
| K (μmol/seedling) | 381 (22.5)a | 432 (23.7)a | 444 (30.9)a | 388 (24.1)a | 395 (31.7)a |
| Mg (μmol/seedling) | 429 (34.6) a | 537 (49.9) a | 442 (31.4)a | 454 (47.3)a | 545 (69.4)a |
| Mg (μmol /g seedling) | 112 (8.0)a | 120. (9.6)a | 95 (3.9)a | 110. (6.8)ab | 139 (12.7)b |

**Table 4: Elemental contents (μmol) in *P. sylvestris* seedlings of Ca, K, Si, and Mg.**
**Values paired with different letters are significantly different ($p < 0.05$) either between ambient and elevated $CO_2$ (left half of table) or between planted treatments (right half of table).**

| Element | Ambient $CO_2$ | Elevated $CO_2$ | Non- planted | Non-mycorrhizal | *Suillus variegatus* | *Piloderma fallax* |
|---|---|---|---|---|---|---|
| Si | 325 (45.6)a | 330 (42.1)a | 142 (12.4)a | 311 (53.7)b | 286 (46.7)b | 383 (57.3)b |
| Ca | 416 (54.5)a | 438 (45.0)a | -246 (17.1)a | 505 (73.5)b | 465 (56.0)b | 326 (42.7)b |
| K | 270 (17.1)a | 273 (27.2)a | 15 (13.3)a | 266 (43.1)b | 262 (18.2)b | 285 (23.8)b |
| Mg | 280 (35.4) a | 377 (52.3) a | -175 (4.84)a | 264 (34.2)b | 313 (49.7)b | 396 (68.5)b |

**Table 5: Total weathering (μmol) losses for Ca, K, Mg, and Si (average/ column). Total weathering = leachate + ΔEC + seedling uptake – nutrient additions. Values paired with different letters are significantly different ($p < 0.05$) either between planted columns with ambient and elevated $CO_2$ (left half of table) or between biological treatments (right half of table).**





**Figure 1: Chitin contents (mean/column) by treatment. Bars that do not share letters are significantly different (p < 0.05).**

**Figure 2: Total chitin content per unit seedling mass (mg/g) for planted treatments by $CO_2$ treatment. Error bars are equivalent to the standard error of the mean in length. Bars that do not share letters are significantly different (p < 0.05).**

**Figure 3: Effects of (a) biological treatment and (b) $CO_2$ treatment (planted columns only) on concentrations of low molecular weight organic acids (LMWOA) in rhizosphere soil solution collected with lysimeters (µmol/L). Values are the average of five sampling occasions. Bars that do not share letters are significantly different (p < 0.05). Capital letters atop each bar refer to total LMWOA.**

**Figure 4: pH of leachate for planted (n=28) and non-planted columns (n=8) on 7 different sampling dates. Error bars are equivalent to the standard error of the mean in length. Pairs of columns topped with an * represent significant differences (p <0.001).**

**Figure 5: Total elemental (µmol) losses for (a) Si, (b) Ca, (c) Mg, and (d) K (average/column). non-plant = non-planted columns. non-myc = non-mycorrhizal columns. *P. fall.* = columns with seedlings colonized by *P. fallax*. *S. var.* = columns with seedlings colonized by *S. variegatus*. Negative values for ΔEC indicate that EC decreased over the course of the experiment.**









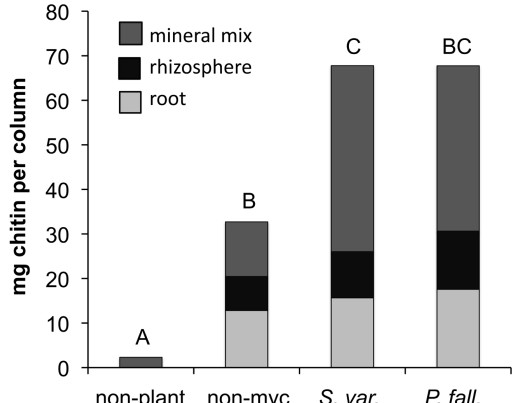

**Figure 1**

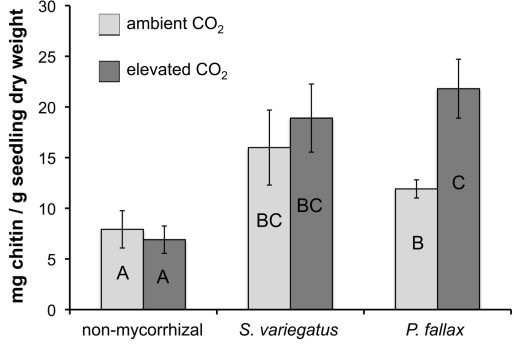

**Figure 2**





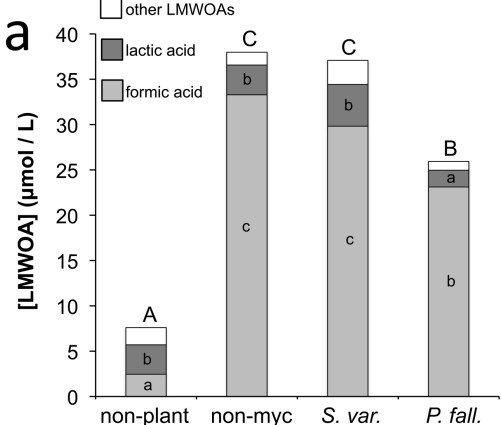

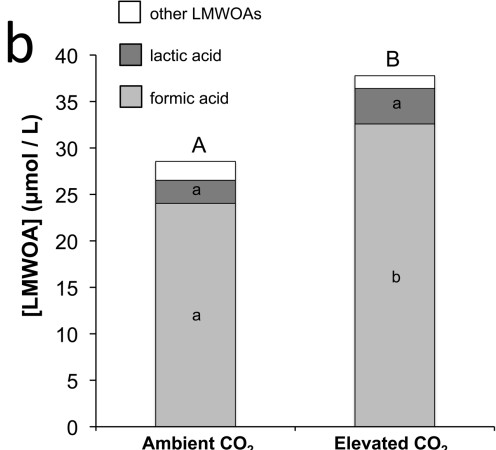

**Figure 3**

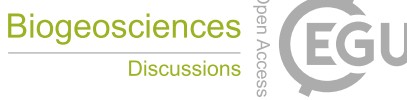

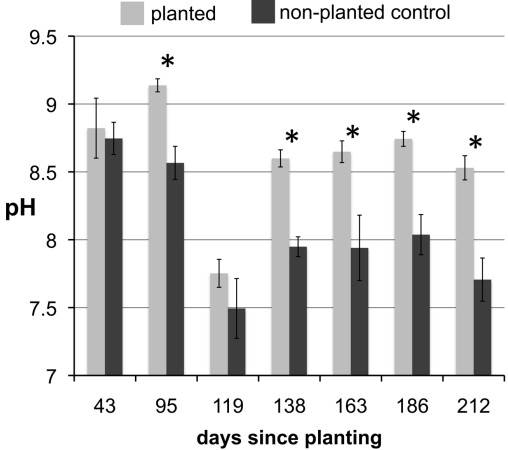

**Figure 4**





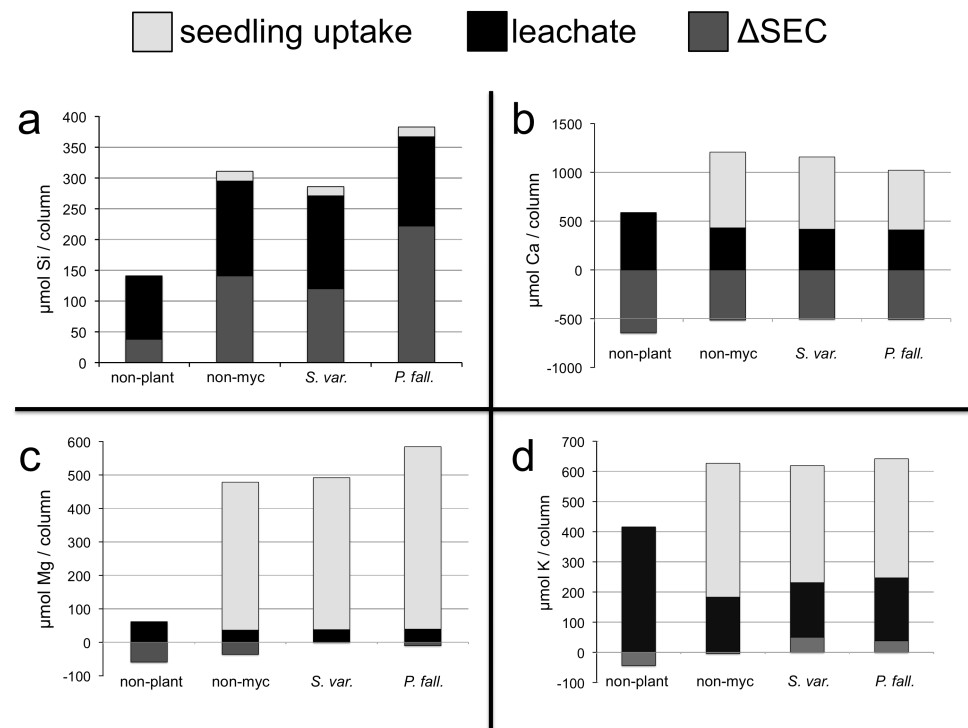

**Figure 5**