# Peer review of "Biological enhancement of mineral weathering by *Pinus sylvestris* seedlings - effects of plants, ectomycorrhizal fungi, and elevated $CO_2$ ."

_Biogeosciences, 2019_

## Short Comment (SC1) · 20 Mar 2019

Kazumichi Fujii

fkazumichi@affrc.go.jp

Received and published: 20 March 2019

This paper has dealt with effects of plant, mycorrhiza, and future climate (elevated CO2) on mineral weathering using seedling column study. The experiment was welldesign and conducted to evaluate effects of mycorrhizal association and CO2 elevation on mineral weathering rates in soil. Discussion on mineral weathering is reasonable, except for one critical issue.

Major issue: I am afraid of underestimation of weathering rate of Si or Si uptake by

plants in this study. Plant Si uptake or Si concentration in plant materials could not be determined only by acid digestion. The majority of Si will remain as residue in acid solution and the residue need to be included as Si uptake by plants. This problem has already been raised by VC Farmer. The Si solubility increases with increasing pH. This can explain the greater Si weathering in the planted column with the higher pH. If actual plant Si uptake is considered, Si weathering in planted column will be further greater.

Minor issue: The dominant low molecular weight organic acids detected in this study are formic, lactic, and acetic acids. They are relatively weak agents for mineral weathering, compared to chelating oxalic and citric acids that are well-known to contribute to mineral weathering in podzol E horizon. Is there any idea to explain the difference of root exudates between the previous studies (e.g., Ahonen-Jonnarth et al., 2000) and this study? Possible reasons may be high availability of P in added solution, low Al levels in soil solution, and precipitation of Ca oxalate. Reasoning is beneficial to generalize the result obtained. The function of formic acid, the dominant organic acid in this study, by roots needs to be further discussed.

Most data are presented on a basis of column. I wonder whether the results could be transformed and presented in a generalized manner to compare with the previous studies.

Reference Ahonen-Jonnarth, U., Van Hees, P. A., LUNDSTRÖM, U. S., & Finlay, R. D. (2000). Organic acids produced by mycorrhizal Pinus sylvestris exposed to elevated aluminium and heavy metal concentrations. The New Phytologist, 146(3), 557-567.

---

## Referee Comment (RC2) · Anonymous Referee #2 · 24 Apr 2019

The ms describes an experiment to assess the relative influence of plants and two associated ectomycorrhizal fungi on weathering budgets. By executing the experiment under both ambient and elevated CO2 the authors also wanted to address the issue whether elevated CO2 would affect weathering rates. In the experiment appropriate controls without plants were included. It is a pity that the supposedly non-mycorrhizal control was (partly) mycorrhizal. While molecular methods could have been used to identify that fungus, I would not think this is a major problem, as the paper does not make any claim about ectomycorrhizal fungal enhancement of weathering rates. How-

ever, it may be preferable to refer to the treatment as the control or non-inoculated treatment rather than to the non-mycorrhizal treatment. While in my view the design of the experiment is OK, I found interpretation of the data more complicated. Part of the data certainly are in support of (or are at least consistent with) a biotic mechanism that enhances weathering, for which reduced transport limitation is proposed as the driving factor. However, in order to focus on the weathering story some inconvenient facts do not receive the attention that they deserve in the view of this reviewer. Negative weathering losses (Table 5) for Ca (246 $\mu$mol) and Mg (175 $\mu$mol) in the unplanted controls do not receive much attention. The authors refer to this negative value as a missing sink and suggest that the cause may be sought in what happened in the three-week flushing phase before planting. They also suggest that, were this explanation correct, the effect would be similar for both planted and unplanted treatments, and hence would not affect the calculations. While that may be true, that explanation fails to provide any suggestion why that missing sink is so different for K (no missing sink at all) and Mg. This reviewer would like to know better how likely the flushing effect was. If that effect was major, one would expect also relatively large leaching losses in the first leachates compared to the leachates that were collected towards the end of the experiment (as the missing sink implies leaching losses in the period that there were no measurements undertaken). It may then be interesting to connect these to the observed leaching losses (Table 2) for Ca and Mg that differ almost an order of magnitude. As the ms states that the eleven leachates were all analysed separately (p. 4, l. 20-22) I think that the temporal pattern for leaching losses would allow a better evaluation of the arguments for the missing sink. In my view the crucial table 3 (with $\Delta$EC) demands more reflection; and providing data on the time course of $\Delta$EC would be very helpful. Whereas many previous studies have shown a large role for ectomycorrhizal fungi (certainly members of the Boletales like Suillus, Paxillus and Rhizopogon) in mineral weathering and a small role for non-mycorrhizal seedlings in weathering, this study does not find no evidence for an ectomycorrhizal fungal role (despite the title), nor does it find evidence for the production of di- and tricarboxylic acid production by

ectomycorrhizal fungi. The discussion on that discrepancy is (too) short in my view. Also the lack of effect of elevated CO2 on the weathering budget (even though it increased allocation belowground and production of LMWOA) is somewhat curious in view of earlier (presumed) knowledge on the role of ectomycorrhizal fungi in weathering. Based on these results the authors of this ms conclude that production of organic ligands (the anions of these LMWOAs) are not the main mechanism for weathering. As they also did not find lowering of pH, they also state that that hypothesis (acidification) can be refuted as a main mechanism for weathering. The ms lists two further mechanisms, but while physical disruption is mentioned, the data are not discussed in relation to this theory. The authors then suggest that alleviation of transport limitation is the driving mechanism. I am not sure whether I understand this hypothesis correctly. It seems that the concentration in the soil solution is higher than plant demand (as leaching losses are substantial compared to plant uptake), so why (to put it in anthropomorphic terms) would plants increase weathering rates way beyond their demand? What I found somewhat surprising that no attention is given to the possibility of (some) weathering as a consequence of autotrophic respiration (by roots and ectomycorrhizal fungi). Root respiration has been proposed as a major weathering agent; and while the authors may disagree with that point of view, I think it is fair that they discuss this possibility. Considering the likely large difference in contribution by heterotrophic respiration (based on low fungal biomass in Fig. 1) and autotrophic respiration, I think the issue merits more attention. While the causes for the high pH of the leachates remains unknown, one could well imagine that increased CO2 production would have lowered leachate pH (Figure 4).

---

## Author Comment (AC2) · 31 May 2019

Referee comment: The ms describes an experiment to assess the relative influence of plants and two associated ectomycorrhizal fungi on weathering budgets. By executing the experiment under both ambient and elevated CO2 the authors also wanted to address the issue whether elevated CO2 would affect weathering rates. In the experiment appropriate controls without plants were included. It is a pity that the supposedly non-mycorrhizal control was (partly) mycorrhizal. While molecular methods could have been used to identify that fungus, I would not think this is a major problem, as the

paper does not make any claim about ectomycorrhizal fungal enhancement of weathering rates. However, it may be preferable to refer to the treatment as the control or non-inoculated treatment rather than to the non-mycorrhizal treatment.

Author Response: We have now, throughout the manuscript, followed the referee's suggestion and changed all occurrence of non-mycorrhizal to non-inoculated as well as mycorrhizal to inoculated.

Referee comment: While in my view the design of the experiment is OK, I found interpretation of the data more complicated. Part of the data certainly are in support of (or are at least consistent with) a biotic mechanism that enhances weathering, for which reduced transport limitation is proposed as the driving factor. However, in order to focus on the weathering story some inconvenient facts do not receive the attention that they deserve in the view of this reviewer. Negative weathering losses (Table 5) for Ca (246 $\mu$mol) and Mg (175 $\mu$mol) in the unplanted controls do not receive much attention. The authors refer to this negative value as a missing sink and suggest that the cause may be sought in what happened in the three-week flushing phase before planting. They also suggest that, were this explanation correct, the effect would be similar for both planted and unplanted treatments, and hence would not affect the calculations. While that may be true, that explanation fails to provide any suggestion why that missing sink is so different for K (no missing sink at all) and Mg.

Author Response: We think there was also a missing sink for K, likely larger than for Mg. That overall weathering was larger for K, and not negative in the unplanted treatment is obscured by the fact that more K was weathered during the course of the experiment so the amount potentially lost in the beginning did not make the total calculated weathering negative. The total amount of K weathered, noted in table 5, is only very slightly positive. The mineral mix employed, (50% quartz , 28% oligoclase, 18% microcline, 1.8% hornblende, 0.9% vermiculite, and 0.9% biotite), may have resulted in more of a spike in Mg losses than K, but it is difficult to speculate on this. This information is discussed at the start of section 4.2.

Referee comment: This reviewer would like to know better how likely the flushing effect was. If that effect was major, one would expect also relatively large leaching losses in the first leachates compared to the leachates that were collected towards the end of the experiment (as the missing sink implies leaching losses in the period that there were no measurements undertaken). It may then be interesting to connect these to the observed leaching losses (Table 2) for Ca and Mg that differ almost an order of magnitude. As the ms states that the eleven leachates were all analysed separately (p. 4, l. 20-22) I think that the temporal pattern for leaching losses would allow a better evaluation of the arguments for the missing sink. In my view the crucial table 3 (with $\Delta$EC) demands more reflection; and providing data on the time course of $\Delta$EC would be very helpful.

Author Response:

Cumulative elemental losses (mean $\mu$mol lost per column), averaged across all columns, in leachate over time.

We could not take measurements of $\Delta$EC. We do not observe a significant temporal trend in leachate losses when we examine leachate losses over time (see figure above). However, unfortunately, leachate losses were not collected for the time period during which the columns were not planted. As added material consisted of freshly ground primary minerals, it is expected that there are many small edges which may weather rapidly upon contact with water. We acid-washed the quartz sand to reduce this spike of Si and we, indeed, observed an increase in salt–extractable Si over the time course of the experiment, as we would expect. We did not perform this acid wash with the other 50% of the mineral mix (28% oligoclase, 18% microcline, 1.8% hornblende, 0.9% vermiculite, and 0.9% biotite), because we were concerned that we would lose too much base cations to non-stoichiometric dissolution of surface bound cations, but, knowing that there would likely be a flush of base cations when the columns were packed and water flowed though them we settled on this initial 3 weeks of equilibration, during which the columns were watered at a higher rate than during the experimental

period, as a solution to reduce this pulse effect of mineral dissolution. We should have collected some of the mineral mix from the columns at exactly the time of planting, but we neglected to do so; we instead saved material from the original mineral mix. When we measured ammonium-acetate exchangeable elements on the final and initial mineral mix, we attempted to account for this 3 week phase by performing 3 sequential rinses of the mineral mix, using solution volumes that corresponded to the relative solution:mineral mix volume used during the equilibration phase, assuming that this would be sufficient to capture the spike in mineral weathering of the freshly ground mineral surfaces. However, in hindsight, this was insufficient to replicate the equilibration phase, as the vertical percolation of the columns allowed mineral elements to progressively migrate downwards, reducing solute concentrations at the mineral surfaces, creating a nearly counter-current flow system. The shaking setup we used during our attempt to replicate this phase did not achieve the same extraction efficiency. We acknowledge that the other potential sink for these cations could be secondary precipitates that we did not extract at harvest and attempted to predict which ones might have been expected to form given our solution chemistry and have now discussed the effect the occurrence of precipitates would have had on the weathering budget.

Referee comment: Whereas many previous studies have shown a large role for ectomycorrhizal fungi (certainly members of the Boletales like Suillus, Paxillus and Rhizopogon) in mineral weathering and a small role for non-mycorrhizal seedlings in weathering, this study does not find no evidence for an ectomycorrhizal fungal role (despite the title),...

Author Response: We are not sure if the reviewer used a double negative intentionally, as the sentence makes sense with and without the second "no". We therefore attempt to address both possible comments. Many studies have found a stimulatory effect on mineral weathering as we discussed in the introduction. However, a number of studies, in both field and microcosm settings, have failed to find an effect of ectomycorrhizal fungi on mineral weathering (Smits et al., 2014; Calvaruso et al., 2010; Koele et al.,

2014). In our study there was some indication that there could have been some elevated weathering activity in the inoculated treatments, particularly for P. fallax. We have added text to the discussion (4.3 Biotic enhancement of weathering) to examine this evidence as well as potential reasons why our mycorrhizal inoculation treatments may have failed to result in a significant mycorrhizal stimulation of mineral weathering.

Changes in manuscript: "There was, however, limited evidence for ectomycorrhizal stimulation of mineral weathering. Ammonium-acetate extractable K and Mg in the mineral mix were greater in the mycorrhizal treatments, and P. fallax appeared to solubilize more Si and Mg (23% and 27% greater mobilization, respectively, than non-mycorrhizal treatments, though not significantly) and take up significantly more Mg than either S. variegatus or the non-mycorrhizal treatment. Overall though, ectomycorrhizal treatments did not exhibit significantly greater weathering rates. It is possible that had our mycorrhizal treatments been more consistently colonized, and had the non-mycorrhizal treatment remained entirely free of mycorrhiza, we might have seen a clearer effect of mycorrhizal treatment on mineral weathering. However, half of the mycorrhizal treatment seedlings, and none of the non-inoculated seedlings, were heavily or moderately colonized at harvest. Further, when we examine seedling chitin content as an explanatory factor, there is no significant relationship between ectomycorrhizal colonization and the weathering fluxes of any element. Another reason for the lack of mycorrhizal stimulation of mineral weathering may be the limited rooting volume to which the seedlings were confined. The upper half of the columns had very dense rooting, and thus the ability for ectomycorrhizal fungi to greatly increase the exploitable soil volume of a plant was not appreciably incorporated into this experimental arena."

Calvaruso, C., Turpault, M.P., Uroz, S., Leclerc, E., Kies, A., Frey-Klett, P.: Laccaria bicolor S238N improves Scots pine mineral nutrition by increasing root nutrient uptake from soil minerals but does not increase mineral weathering, Plant and Soil, 328, 145–154, 2010.

Koele, N., Dickie, I.A., Blum, J.D., Gleason, J.D., de Graaf, L.: Ecological significance

of mineral weathering in ectomycorrhizal and arbuscular mycorrhizal ecosystems from a field-based comparison, Soil Biology & Biochemistry, 69, 63 – 70, 2014.

Smits, M.M., Johansson, L., Wallander, H.: Soil fungi appear to have a retarding rather than a stimulating role on soil apatite weathering, Plant and Soil, 385, 217–228, 2014.

Referee comment: . . . nor does it find evidence for the production of di- and tricarboxylic acid production by ectomycorrhizal fungi. The discussion on that discrepancy is (too) short in my view.

Author Response: Our findings of relatively low concentrations of citric and oxalic acid (compared to monocarboxylic lactic, acetic, and formic acids, seem to be in line with the majority of studies that have examined low molecular weight organic acid concentrations in non-sterile settings. In addition, the lack of a mycorrhizal efffect on low-molecular weight organic acid production, when seedlings were not subjected to aluminum and heavy metal exposure seems also to be in line with many, if not most, other studies under similar conditions. The literature on ectomycorrhizal stimulation of soil solution organic acid concentrations is ambiguous and the "clear picture" that EMF promote significant oxalic and citric acid production is primarily derived from pure-culture experiments where microbial degradation is not occurring, and may thus, be difficult to replicate in non-axenic settings.

Changes in manuscript: We have added to the discussion (4.1 Growth and organic acid concentrations) to place our findings in a context in light of other studies and the particular study system we employed. "While EMF are often mentioned in the literature to produce significant amounts of LMWOA's, our findings seem to fall in line with the majority of studies examining the EMF role in LMWOA production which fail to find an increase in LMWOA production when comparing EMF and non-EMF seedlings (van Scholl et al., 2006; van Hees et al. 2005). However, many of these studies do find that EMF significantly alter the composition of LMWOA's produced, particularly increasing oxalic acid concentrations (van Scholl, 2006; van Hees et al. 2006, Ahonen-Jonnarth,

2000), which we did not. However, a number of studies in ectomycorrhizal systems have also observed higher soil solution concentrations of formic, acetic, and lactic acids than of the weathering promoting citric and oxalic acids (Strobel et al., 2001; van Hees et al., 2006; van Hees et al., 2002; Ray and Adholeya, 2009). Studies that find particularly high concentrations of oxalic acid, are typically from pure culture studies with high nitrogen availability (Rosling 2009), where rapid microbial degradation of LMWOAs is excluded. Despite the fact that our rhizosphere lysimeters were placed in the area of highest root density it is possible that the organic acid profile we observed is primarily the product of microbial activity, following partial decomposition of plant exudates and other SOM. It is, however, interesting to note that the planted treatments all had far higher LMWOA concentrations, and that LMWOA concentrations were correlated with seedling biomass."

Referee comment: Also the lack of effect of elevated CO2 on the weathering budget (even though it in- creased allocation belowground and production of LMWOA) is somewhat curious in view of earlier (presumed) knowledge on the role of ectomycorrhizal fungi in weathering.

Author Response: We also expected to observe greater mineral weathering with CO2 addition. Root biomass was not significantly enhanced by CO2 addition, and while LMWOA concentrations were higher, LMWOA concentrations per unit seedling mass were not higher. As discussed above, and now, in response to the reviewers' comments, in greater detail in the manuscript, we did not observe LMWOA concentrations that would be expected to appreciably stimulate mineral weathering.

Referee comment: Based on these results the authors of this ms conclude that production of organic ligands (the anions of these LMWOAs) are not the main mechanism for weathering. As they also did not find lowering of pH, they also state that that hypothesis (acidifi- cation) can be refuted as a main mechanism for weathering. The ms lists two further mechanisms, but while physical disruption is mentioned, the data are not discussed in relation to this theory.

Author Response: We observed consistently and significantly elevated pH in the column leachate from planted controls, and this would not be expected if proton exudation were the predominant mechanism by which seedlings stimulating mineral weathering. We have discounted physical disruption as a major contributing process to the biological stimulation of weathering due to our use of fresh ground primary mineral without pre-existing fissures and cracks for hyphae to exploit and the timespan of the experiment, which have excluded the potential for significant increase of available mineral surface area by biomechanical forcing (Pawlik et al., 2016).

Changes in manuscript: We have added a sentence in the discusssion to address this. "We have discounted physical disruption as a major contributing process to the biological stimulation of weathering due to our use of fresh ground primary mineral without pre-existing fissures and cracks for hyphae to exploit and the timespan of the experiment, which have excluded the potential for significant increase of available mineral surface area by biomechanical forcing (Pawlik et al., 2016). "

Pawlik L, Phillips JD, Šamonil P, Roots, rock, and regolith: Biomechanical and biochemical weathering by trees and its impact on hillslopes—A critical literature review. Earth-Science Reviews 159 (2016) 142–159

Referee comment: The authors then suggest that alleviation of transport limitation is the driving mechanism. I am not sure whether I understand this hypothesis correctly. It seems that the concentration in the soil solution is higher than plant demand (as leaching losses are substantial compared to plant uptake), so why (to put it in anthropomorphic terms) would plants increase weathering rates way beyond their demand?

Author Response: We do not assert that plants are actively weathering in response to a stimulus, or actively expending resources for the "purpose" of stimulating mineral weathering, but rather, that, in our system, the uptake of nutrients from the solution around primary minerals releases a brake on weathering rates, by reducing re-adsorption of weathering products onto mineral surfaces, increasing mineral weathering rates. We calculated the portion of all elements entering solution that were taken up by seedlings to examine the effect of uptake on solution concentrations and observed that were weathering rates not in equilibrium with surficial solution, then seedling uptake would have reduced solution concentrations by 54 - 91%.

Referee comment: What I found somewhat surprising that no attention is given to the possibility of (some) weathering as a consequence of autotrophic respiration (by roots and ectomycorrhizal fungi). Root respiration has been proposed as a major weathering agent; and while the authors may disagree with that point of view, I think it is fair that they discuss this possibility. Considering the likely large difference in contribution by heterotrophic respiration (based on low fungal biomass in Fig. 1) and autotrophic respiration, I think the issue merits more attention. While the causes for the high pH of the leachates remains unknown, one could well imagine that increased $CO_2$ production would have lowered leachate pH (Figure 4).

Author Response: The increased pH in the column leachate from planted controls would seem to negate the potential for root respiration to be a significant driver of the enhanced mineral weathering observed in the planted treatments.

Please also note the supplement to this comment:
https://www.biogeosciences-discuss.net/bg-2019-46/bg-2019-46-AC2-supplement.pdf

[Figure]

Cumulative elemental losses (mean μmol lost per column), averaged across all columns, in leachate over time.

**Fig. 1.**

---

## Author Comment (AC1)

[revised manuscript text omitted]